# Overexpression of GhKTI12 Enhances Seed Yield and Biomass Production in Nicotiana Tabacum

**DOI:** 10.3390/genes13030426

**Published:** 2022-02-25

**Authors:** Aye Aye Myat, Yu Zhou, Yuan Gao, Xiang Zhao, Chengzhen Liang, Muhammad Ali Abid, Peilin Wang, Umar Akram, Mubashir Abbas, Muhammad Askari, Sandui Guo, Rui Zhang, Zhigang Meng

**Affiliations:** 1Biotechnology Research Institute, Chinese Academy of Agricultural Sciences, Beijing 100081, China; aamyat19may@gmail.com (A.A.M.); 82101182062@caas.cn (Y.Z.); plateau1995@163.com (Y.G.); zx19921301161@163.com (X.Z.); liangchengzhen@caas.cn (C.L.); abid@caas.cn (M.A.A.); wangpeilin19@126.com (P.W.); umar02kwl@gmail.com (U.A.); mubashirabbas3164@yahoo.com (M.A.); askaricaas5@yahoo.com (M.A.); guosandui@caas.cn (S.G.); zhangrui@caas.cn (R.Z.); 2Institute of Plant Breeding and Biotechnology, MNS—University of Agriculture, Multan 60000, Pakistan

**Keywords:** *GhKTI12*, seed yield, biomass, transgenic tobacco

## Abstract

Crop molecular breeding primarily focuses on increasing the trait of plant yield. An elongator-associated protein, *KTI12*, is closely associated with plant biomass and yield. *KTI12* is involved in developmental processes of most organs, including the leaf, root, flower, and seed, through regulating cell division and differentiation. Previous work has shown that in upland cotton (*Gossypium hirsutum*), *GhKTI12* regulates plant height, flowering, and tolerance to salt and drought stress. However, little is known about the molecular regulation mechanism of *GhKTI12* in plant developmental processes. In this study, we identified the main *GhKTI12* (*Gh_D02G144400*) gene and transformed it into tobacco (*Nicotonia tabacum* cv NC89). From seven transgenic lines, we obtained three (OE5, OE6 and OE8) with high expression of *GhKTI12*; compared with wild type plants, these three lines exhibited larger plant size, later flowering, and higher seed yield. Microscopic observation revealed that the number of leaf epidermal cells and stem parenchyma cells was increased by ~55%. Biochemical analysis showed that chlorophyll content and starch accumulation were significantly increased in younger leaves at the top canopy of transgenic plants, which may contribute to improved photosynthetic rate and, in turn, increased seed yield. To understand the molecular mechanism of *GhKTI12* in transgenic plants development, two lines (OE6 and OE8) with higher expression levels of *GhKTI12* were used as representative plants to conduct RNA-seq analysis. Through transcriptome analysis of the plant’s shoot apical meristematic tissue of these two lines, we identified 518 upregulated genes and 406 downregulated genes common to both overexpression lines. A large number of cellular component genes associated with cell division and differentiation, such as *RD21*, *TET8*, *KTN80*, *AOX1*, *AOX2*, *CP1*, and *KIC*, were found to be upregulated, and genes showing the most downregulation included MADS-box genes related to flowering time, such as *MADS6*, *AP1*, *AP3*, *AGL8*, *AGL6*, *SEP1*, and *SEP2*. Downregulation of these genes caused delayed flowering time and longer vegetative stage during development. Combined with the upregulation of the yield-related gene *RD21*, the *GhKTI12* transgenic plants could produce a higher seed yield. We here show that the overexpression of *GhKTI12* could positively improve key agronomic traits in tobacco by regulating cell proliferation, photosynthesis, and organ development, and suggest that homologs of *GhKTI12* may also be important in the genetic improvement of other crop plants.

## 1. Introduction

The *KTI* (Killer Toxin Insensitive or Kluyveromyces lactis Toxin Insensitive) gene encodes an elongator-associated protein. Three *KTI* genes (*KTI11*, *KTI12*, and *KTI13*) were initially isolated from *Saccharomyces cerevisiae* [1,2,3,4]. *KTI12* is a highly conserved ATPase, and is necessary for the tRNA-modification activity of the eukaryotic Elongator complex [3,5]. *KTI12* homologs have been identified in many plants, such as *Arabidopsis thaliana*, *Oryza sativa*, *Zea mays*, *Brassica napus*, *Populus* [6].

Yeast KTI12 and plant orthologs contain an N-terminal P-loop motif [7] and a putative calmodulin binding domain (CBD) [8]. Loss of CBD on the C-terminal of KTI12 results in zymocin resistance in yeast [9,10]. The P-loop motif is the primary functional area of the KTI12 protein and contains O-phospho-seryl-tRNA-kinase (PSTK) activity. It is conserved in all known KTI12 homologs and is an essential component for elongator-dependent tRNA modification [9]. Like the CBD terminal, loss of the P-loop function in KTI12 yeast mutants results in zymocin resistance [11]. In plants, the P-loop region in KTI12 regulates leaf morphogenesis via wobble uridine modification [6]. A recent study revealed that deficiencies in tRNA wobble modification alter leaf development through increasing leaf epidermal cells in the Atelp3 mutant [12].

Plant KTI12 homologs are localized to the nucleus [3] and regulate gene transcription through interaction with elongator, a complex containing two sub-complexes and six subunits (ELP1-ELP6) that are highly conserved in eukaryotic organisms including yeast, humans, and Arabidopsis [13]. The core subcomplex of an elongator consists of ELP1, ELP2 and ELP3. KTI12 can affect tRNA modification to regulate plant development through interaction with ELP. Mutation of *AtKTI12* and *AtELP1* in Arabidopsis were shown to reduce organ growth and early flowering, respectively [14].

*DEFORMED ROOTS AND LEAVES1* (*DRL1*) is a *KTI12* homolog that can modulate cell division and differentiation to regulate meristem activity and organ growth in Arabidopsis [4,8]. *AtDRL1/KTI12* mutants in Arabidopsis have defective shoot and root meristems [4,15]. Complementation experiments of yeast *KTI12* with Arabidopsis and rice drl1 mutant showed rescued growth retardation [8].However, *ELP* mutants in Arabidopsis show pleiotropic phenotypes including narrow leaves, enlargement of hypocotyl, retarded primary root growth, decreased seed germination, delayed flowering time, and reduced apical dominance [4,9,16,17]. The comprehensive morphological and molecular characterization of DRL1 and elongator gene mutants demonstrate that KTI12 and DRL1 have protein–protein interactions with the elongator in yeast and Arabidopsis, respectively.

Tobacco (*Nicotiana tabacum* L.) is an economically important plant, and the variety NC89 has been widely used as a model for gene functional analysis. In a previous study, we screened functional genes related to plant architecture, abiotic stress tolerance, and yield in cotton using RNA-seq data and the virus-induced gene silencing (VIGS) method. We found that *KTI12* affects plant height, flowering, and tolerance to salt and drought stress in upland cotton. There are two homologous genes, *Gh_A03G123100* and *Gh_D02G144400*, in the cotton genome. From gene expression profiles and stress-induced characteristics, we found that *Gh_D02G144400* is likely the gene primarily responsible for *KTI12* function. However, very few studies have functionally characterized the *GhKTI12* gene. In this study, we transformed *GhKTI12* (*Gh_D02G144400*) into the tobacco variety NC89. Transgenic tobacco plants with high expression of *GhKTI12* were identified and used to analyze gene function based on phenotypes. The results of this study provide a better understanding of *KTI12* gene function in key plant traits and lay an experimental foundation for genetic crop improvement.

## 2. Materials and Methods

### 2.1. Plant Materials and Growth Conditions

Tobacco seeds were provided by Biotechnology Research Institute, Chinese Academy of Agricultural Sciences Beijing, China. Seeds were germinated and grown in ½ Murashige and Skoog (MS) medium Petri plates and incubated at 25 °C in a growth chamber with a 16 h photoperiod for 15 days. Precultured leaves were used as explants for Agrobacterium tumefaciens-mediated genetic transformation [18].

### 2.2. Plant Transformation and Identification of Transgenic Tobacco

The overexpression vector pCAMBIA2300-35S-GhKTI12 was constructed by inserting the coding sequence of *GhKTI12* into the empty vector pCAMBIA2300-35S between *Kpn*I and *Sma*I sites. The pCAMBIA2300-35S plasmid was provided by Biotechnology Research Institute, Chinese Academy of Agricultural Sciences in Beijing, China. The overexpression vector was transformed into Agrobacterium strain GV3101. Transgenic tobacco plants were created by the Agrobacterium-mediated leaf disc method [19]. After obtaining kanamycin-resistant plants, DNA was extracted from the transgenic and wild type (WT) plants by using NuClean Plant Genome DNA Kit (ComWin Biotech Co., Ltd., Jiangsu, China) to identify the target gene via PCR amplification. The PCR Mix was 2 × Phanta Max Super-Fidelity DNA Polymerase (Vazyme Biotech Co., Ltd., Nanjing, China). Primers shown in Appendix A were used to amplify a 912-bp fragment of the *GhKTI12* gene. The PCR program was as follows: pre-denaturation at 95 °C for 3 min; 30 cycles of amplification (95 °C for 30 s, 56 °C for 45 s, 72 °C for 45 s); followed by 5 min at 72 °C. T2 homozygous lines were generated to conduct further experiments.

### 2.3. Total RNA Isolation and qRT-PCR Analysis

For transcript expression analysis, RNA from different tissue samples was extracted using a FastPure Plant Total RNA Isolation Kit (Vazyme Biotech Co., Ltd., Nanjing, China). First-strand cDNA synthesis was conducted with HiScript II Q RT SuperMix for qPCR (+gDNA wiper) kit (Vazyme Biotech Co., Ltd., Nanjing, China), following the manufacturer’s instructions. The 2^−ΔΔCT^ method was used to calculate relative expression level [20], and *NtACT7* was used as the internal reference gene. Primers are shown in Appendix A. All qRT-PCR experiments were conducted with three technical replicates.

### 2.4. Collection of Morphological Data of T2 Transgenic Tobacco

Morphological traits of transgenic and WT plants were recorded for the T2 generation including leaf size, plant height, number of leaves, number of internodes, flowering time, capsule number, and seed yield at flowering time. The fresh and dry weight of vegetative tissues (leaves, stems, and roots) were measured. Transgenic plants were separately harvested and dried at 70 °C after harvesting.

### 2.5. Chlorophyll Content Determination

Chlorophyll content in transgenic and WT plants was determined as described by Parry (2014) [21]. Briefly, 0.1 g of plant tissue was homogenized in 80% acetone and incubated in the dark for 6 h. The homogenate was centrifuged at 10,000× *g* for 10 min. Absorbance of the supernatant was measured at 649 and 665 nm with a Spark plate reader (Tecan, Männedorf, Switzerland).

### 2.6. Histological Analysis

Leaf tissue and stem internode samples were collected from 45- to 60-day-old GhKTI12 transgenic and WT plants, then fixed in a solution of 3:1 ethanol: acetic anhydride with a drop of Tween-20 for 1 h [22]. Samples were dehydrated in a series of ethanol solutions (70%, 80%, 85%, 90%, 95%, and 100% anhydrous ethanol) and destained in a series of Technovit 7100 (Heraeus, Hanau, Germany) solutions (3:1 ethanol:xylene, 1:1 ethanol:xylene, 1:3 ethanol:xylene, and pure xylene). Each leaf sample was placed in a mold or microcentrifuge tube and Technovit solution. The resin polymerized overnight at 37 °C in a closed environment. Tissue sections were cut with a glass knife, fixed on a glass slide, and stained with 0.05% Toluidine Blue O (Sigma-Aldrich Pty Ltd., Darmstadt, Germany). Microscopic observation of palisade cells and culm parenchyma cells was conducted using a digitized camera with differential interference contrast optics on Laser Confocal Zeiss LSM700 (Carl Zeiss Ltd., Oberkochen, Germany).

### 2.7. Determination of Sucrose and Starch Content

For sucrose and starch content analysis, leaves were harvested from different positions (5th, 6th, 7th, 8th, 9th, and 10th) of three plants for the WT and each of the *GhKTI12* overexpressing lines at the juvenile vegetative stage (approximately 60 days after sowing). A total of 18 leaf samples of three individual plant were selected from each transgenic line. Starch and sucrose content were measured using the β-Amylase Assay Kit and Sucrose Assay Kit (Beijing Solarbio Science & Technology Co., Ltd. Beijing, China) according to the manufacturer protocols.

### 2.8. RNA-Seq and Data Analysis

Total RNA was extracted from the shoot apical meristem of 40-day-old plants with three samples for each line (OE-6, OE-8, and WT) using the RNeasy Plant Mini Kit (Qiagen GmbH, Hilden, Germany). Libraries for RNA-seq analysis were constructed from 3 mg of total RNA per sample using the RNA Library Prep Kit for Illumina (New England Biolabs, Ipswich, MA, USA). Constructed libraries used Illumina high throughput sequencing platform NovaSeq 6000 for sequencing. Differentially expressed genes (DEGs) were identified and gene ontology (GO) enrichment analysis was performed using TBtools [23] and agriGO v2.0 [24]. DEGs with adjusted *p*-value < 0.05 were considered significant. The expression profile of DEGs were visualized with HemI (Heatmap Illustrator, v1.0) [25].

### 2.9. Statistical Analysis for Physiological and Biochemical Experiment

Data were analyzed using SPSS 20.0. Means were compared within transgenic and WT groups using Tukey’s test at the 5% and 1% level. All of the morphological data for T2 transgenic progeny are presented as the mean ± standard error and *p*-value. Differences between transgenic lines and WT were considered if *p* < 0.05. Significantly, differences between transgenic lines and WT were considered if *p* < 0.01.

## 3. Results

### 3.1. Molecular Character of GhKTI12 and Identification of Transgenic Tobacco

There are two *GhKTI12* homologs in upland cotton, and *Gh_D02G144400* is the main functional gene. The full-length coding sequence of *GhKTI12* (*Gh_D02G144400*) is 912 bp and encodes a protein of 303 amino acids in length with a molecular weight of 34 kDa and an isoelectric point of 8.9 (Appendix A). Hydropathy analysis showed that *GhKTI12* contains no transmembrane domain, indicating that it is not a transmembrane protein (Appendix A). Subcellular localization analysis revealed that *GhKTI12* is localized to the nucleus (Appendix A). The predicted secondary structure includes α-helix H bond residues, extra coils, and turn coils (Appendix A).

To elucidate the phylogenetic relationship of KTI12 orthologs among plants, a neighbor-joining tree was constructed using KTI12 amino acid sequences from 16 plant species: Arabidopsis (*Arabidopsis thaliana*), rice ( *Oryza sativa* subsp. japonica and indica), maize (*Zea mays*),upland cotton, diploid cotton (*Gossypium raimondii*), island cotton (*Gossypium barbadense*), *Gossypium arboretum*, tobacco, arabica (*Coffea arabica*), white poplar (*Populus tomentosa* and *Populus alba*), Leprosy tree (*Jatropha curcas*), sesame (*Sesamum indicum*), potato (*Solanum tuberosum*), and rape (*Brassica campestris*) (Figure 1a). The phylogenetic tree showed that the GhKTI12 protein was more closely related to dicot species than monocot species, but there was a distant relationship between cotton GhKTI12 with some monocot species such as rice and maize.

After transformation of the *GhKTI12* overexpression vector (Figure 1b) into tobacco variety NC89, seven independent lines were obtained with kanamycin resistance selection and PCR identification (Appendix A). The overexpression lines were named OE-1, OE-4, OE-5, OE-6, OE-8, OE-12, and OE-13. *GhKTI12* expression was verified in all transgenic lines via qRT-PCR (Figure 1c). *GhKTI12* expression was significantly higher in OE-5, OE-6 and OE-8 than in the other lines, and these were, therefore, selected for further study (Appendix A).

### 3.2. Overexpression of GhKTI12 Could Delay Flowering Time and Increase Plant Height

To identify morphological characteristics, we generated T2 generation plants of OE-5, OE-6 and OE-8. The T2 plants were used to evaluate the number of leaves, internode number, flowering date, and plant height in the whole growth stage. While there was no significant difference in morphology between WT and OE lines before the flowering stage in 60-day-old seedlings (Appendix A), the OE plants had a later flowering date, higher number of leaves, higher mainstem, and more internodes than WT plants (Figure 2a–c).

The initial flowering time was 115 days after sowing in the WT, whereas OE-5, OE-6 and OE-8 did not flower until 145 days, 142 days and 143 days after sowing, respectively (Figure 2d). In the flowering stage, the number of plant leaves in OE-6 and OE-8 were significantly increased compared to the WT (Appendix A); WT plants had only 23.7 ± 2 leaves, but OE-5, OE-6 and OE-8 had 32.44 ± 2, 30.55 ± 2 and 30.75 ± 5 leaves, respectively. In addition to increased leaf number in transgenic plants, the plant height significantly increased and more internodes were developed in the stem (Figure 2e and Appendix A). WT plants were 73 ± 5.4 cm in height, while OE-5, OE-6 and OE-8 plants were 106 ± 7.0 cm 101 ± 3.1 cm and 101 ± 4.8 cm, respectively (Figure 2e), an increase of 45%, ~38% and 36% compared to the WT. The number of mainstem internodes was also significantly increased in transgenic plants with an average of 26, while WT is 20 (Appendix A).

In order to determine the mechanism of morphological changes in transgenic plants at the cytological level, we observed the number, size, and longitudinal length of stem epidermal cells. We found that the cell number in the main stem was 55% higher in OE plants compared to the WT, but the cell size was smaller and longitudinal cell length was shorter (Appendix A). The average cell size of OE plants with 64% of WT and cell length only 77% as well. These results suggest that increased plant height in *GhKTI12* OE plants was mainly caused by increased cell proliferation rather than cell expansion in the mainstem internode.

### 3.3. Overexpression of GhKTI12 Could Increase Leaf Size by Regulating Cell Division

All transgenic plants were larger than WT plants at the flowering stage, particularly the size of the leaf in the middle stem position. The average leaf area was ~36% and 34% higher in OE-6 and OE-8, respectively, compared to the WT.

Ten plants from each line were randomly selected for leaf measurements. The length and width were measured for three leaves in different leaf positions (5th, 8th, and 12th). The average leaf length in transgenic plants was 4.83 cm, 4.79 cm, and 4.60 cm at the 5th, 8th and 12th position, respectively. This was ~36%, 41%, and 59% higher than the corresponding leaves from WT plants (Figure 3a). Although leaf width was also increased in OE plants compared to the WT, the differences were not significant (Figure 3b). These results suggest that the overexpression of *GhKTI12* enhanced leaf length.

Plant organ development size was primarily determined by cell proliferation and cell expansion. To analyze the mechanism of leaf size changes in transgenic plants at the cytological level, we observed cell number and size in the third leaf of 45- to 60-day-old plants under the microscope. The average number of epidermis cells was significantly increased and average cell size was decreased in transgenic plants compared to the WT (Figure 3c); cell number was increased by ~55%, a change which was accompanied by reduced cell size of ~18% (Figure 3d). These results indicate that *KTI12* played a positive role in regulating leaf length by increasing cell proliferation rather than cell expansion.

Taken together, our results support that *GhKTI12* is involved in the positive regulation of leaf epidermal cell division and proliferation for leaf area enlargement in transgenic tobacco plants.

### 3.4. Overexpression GhKTI12 Could Increase Plant Biomass and Seed Yield in Tobacco

All of the OE plants were larger than WT plants at the reproductive stage (Appendix A), particularly in the root system (Figure 4a–c), and OE plants additionally developed more capsules (Appendix A). Statistical analyses indicated that the overexpression of GhKTI12 in tobacco significantly increased the biomass in normal growth conditions (Table 1 and Appendix A). Compared to WT plants, the average whole plant fresh and dry weights were increased by 33% and 34% in transgenic plants (OE-5, OE-6 and OE-8), respectively. The root volume of OE-5, OE-6 and OE-8 was also increased to twice the volume of WT plants (Figure 4a–c). The average root fresh and dry weight were increased by 33% and 52% in transgenic lines, respectively (Table 1). These results suggest that overexpression of *GhKTI12* could promote root development to increase root volume.

OE-5, OE-6 and OE-8 also produced more flowers and capsules in the reproductive stage. The average capsule number per ten plants in OE-5, OE-6 and OE-8 was 114, 128 and 142, which contributed to an increase of 35%, 40% and 44% in net seed yield, respectively (Figure 4d,e), compared to WT plants (Appendix A). However, there was no significant difference in the weight of 1000 seeds between lines (Appendix A). These results suggest that the number of capsules per plant played a major role in increasing seed yield.

### 3.5. Overexpression of GhKTI12 Could Affect the Contents of Chlorophyll, Sucrose and Starch in Tobacco Leaves

In OE plants, leaves appeared greener from the middle to the 10th leaf upper positions than those of WT plants, but had a comparable yellow color in the 6th leaf lower position. Chlorophyll contents were increased approximately 12% and 35% at the 6th and 10th position, respectively, in transgenic plants (Figure 5a). Chlorophyll content analysis showed that overexpression of *GhKTI12* could enhance chlorophyll production during leaf development in transgenic tobacco plants.

In order to understand the mechanism of higher yield and the accumulation of photosynthetic products in transgenic plants, sucrose and starch contents were analyzed in the 5th–10th leaves of 60-day-old OE and WT plants. The leaf sucrose content was gradually increased in the 5th to 8th leaf of transgenic plants, and declined to 13 mg/g fresh weight (FW) in transgenic younger leaves at the 9th position, approximately 16% lower than in the 9th leaf of WT plants (Figure 5b). Conversely, the leaf starch content dramatically increased from the 5th to 10th leaf. The average starch content in OE-5, OE-6 and OE-8 leaves was 1.3–1.45 times higher compared to WT plants. There was also a more than fivefold increase in the OE younger leaf (10th position) compared to the older leaf (5th position), and an approximately threefold increase in the WT 10th leaf compared to 5th leaf (Figure 5c). These results suggest that overexpression of *GhKTI12* could increase chlorophyll content and carbohydrate accumulation, promoting plant development and high yield.

### 3.6. Differential Gene Expression Analysis in Tobacco Transcriptome

*KTI12* interacts with elongator complex to regulate gene expression. Previous results revealed that the overexpression of *GhKTI12* could positively affect plant development. However, there was no significant difference in morphology traits between different transgenic lines (OE-5, OE-6 and OE-8). To explore the molecular mechanism of global gene expression changes in transgenic plants, the OE-6 and OE-8 with most high expression of *GhKTI12* gene were selected to do transcriptome analysis. The transcriptome analysis samples were shoot apical meristematic tissue (SAM) of 60-day-old plants which considered to be the tissue with the most vigorous cell division and differentiation. We identified 11,168 differentially expressed genes (DEGs) from the RNA-seq data comparison of OE-6 and OE-8 with WT plants. Among those DEGs, 2677 (24%) were identified as core DEGs (shared by the two transgenic lines; Appendix A). Among the core DEGs, 518 were upregulated and 406 were downregulated (Appendix A).

### 3.7. Transcriptome Changes Associated with Plant Growth and Development Process

Gene enrichment analysis was performed using single species enrichment analysis (SEA) to gain insight into core DEG functions. The 518 core upregulated genes were enriched in 28 GO functional groups; these included six cellular components, 11 biological processes, and 11 molecular functions (Figure 6a, Appendix A). The terms were related to plant growth and development.

Enriched cellular components included the highly significant (*p*-value < 0.05, FDR < 0.001) respiratory chain (GO:0070469), lysosome (GO:0005764), endoplasmic reticulum lumen (GO:0005788), lytic vacuole (GO:0000323), and “microtubule organizing center part” (GO:0044450). Significantly enriched biological processes were related to immune and heat response, namely regulation of systemic acquired resistance (GO:0010112) and response to heat (GO:0009408). Significantly enriched molecular functions were unfolded protein binding (GO:0051082), glutathione transferase activity (GO:0004364), and α-L-fucosidase activity (GO:0004560) (Figure 6a; Appendix A).

From this enrichment analysis of DEGs, we identified nineteen upregulated genes related to the respiratory chain, lysosome-stored cysteine proteases, microtubule-associated protein and α-L-fructosidase activity (Figure 6b). There were ten respiratory chain genes, which included three alternative oxidative (*AOX*) genes (*AOX1*, *AOX2* and *AOX/AOMI*), four cytochrome c-dependent genes (*CYTC-2*, *COX*, *UQCRX/QCR9* and *petA*), two ATPase inhibitor proteins (*At2g27730* and F1F0-ATPase inhibitor protein), and the transmembrane protein TETRASPANIN 8 (*TET8*). There were five lysosome-stored cysteine protease genes: Cathepsin B-like cysteine proteinase 5 (*Cpr-5*), senescence-associated gene (*SAG12-2*), thiol protease SEN102 (*SEN102*), cathepsin B-like cysteine proteinase (*AT4G01610*) and cathepsin F-like cysteine proteinase (*RD21*). The three microtubule-associated proteins (MAPs) were Katanin p80 (*KTN80*), Ca^2+^-binding protein 1(*CP1*), and Ca^2+^ binding protein with one EF-Hand Motif (*KIC*). Additionally, one gene of α-L-fucosidase activity is *FUC1*.

Among these, thirteen significantly upregulated genes (*F1F0*, *TET8*, *CYTC-2*, *COX*, *SAG12-2*, *RD21*, *Cpr-5*, *CP1*, *KIC*, *KTN80*, *AOX2*, *AOMI* and *AOX1A*) were used to validate gene expression patterns in a mixed sample of OE-6 and OE-8 SAM by qRT-PCR. Compared to relative expression levels in the WT mixed samples, all measured genes were significantly upregulated in transgenic samples (Appendix A). These results validate the RNA-seq results in this system and further suggest that *GhKTI12* could regulate some key genes related to plant cellular and organellar development, ultimately affecting the growth and development of the whole plant.

### 3.8. Transcriptome Changes Associated with Flowering Time

Flowering time was significantly delayed in OE plants compared to the WT. To understand changes in gene regulation associated with delayed flowering in transgenic plants, flowering-related genes were identified among core DEGs. There were 12 enriched GO molecular function groups annotated to 406 downregulated DEGs involved in flowering (Figure 7a; Appendix A). The most enriched GO groups were related to RNA polymerase II regulatory sequence-specific DNA binding (Figure 7a).

Among those 406 DEGs, there were 10 significantly downregulated genes related to flower development and belonging to the MADS-box group, namely *AGL6*, *AGL8*, and *AGL14* from the AGAMOUS (AG) family, and *AP1* (*APETALA1*), *AP3* (*APETALA3*), *SEP1* (*SEPALLATA1*), *SEP2* (*SEPALLATA2*), *MADS6*, *TDR4* (a *AGL8* homolog), and *CMB1* (Figure 7b). According to the importance of their regulation in flowering time, eight genes (*AGL6*, *AGL8*, *AP1*, *AP3*, *AGL14*, *SEP.1*, *SEP.2*, *MADS6*) from 10 significantly downregulated genes were selected for expression level quantification in a mixed sample of OE-6 and OE-8 SAM with qRT-PCR. The results show that those eight genes were significantly downregulated in the OE plants compared with the WT (Appendix A). Similarly, some genes related to magnesium binding, terpene synthase activity, and voltage-gated ion channel activity were also slightly downregulated in OE plants (Figure 7a). These results suggest that delayed flowering was caused by the downregulation of flower development-related genes in *GhKTI12* OE plants.

## 4. Discussion

As an essential component for elongator-dependent tRNA modification [9,14], *KTI12* and its homolog, *DRL1*, can positively regulate plant meristem activity and organ growth by modulating both cell division and differentiation in Arabidopsis [4,8]. Mutants for *AtDRL1*, *AtKTI12*, and *AtELP3* showed abnormal leaf shape and smaller leaf size due to a reduction in leaf palisade cell number and increase in cell size [4,8,9,12,26,27,28]. However, the overexpression of *KTI12* could prolong the vegetative growth stage and promote a larger plant size in Arabidopsis and other plants [29,30].

In this study, OE-5, OE-6 and OE-8 developed in a longer vegetative stage and had larger leaves, increased plant height, larger root systems, and a higher yield. Biochemical analysis revealed that transgenic lines contained more chlorophyll and accumulated more sucrose and starch in functional leaves. In order to explore the underlying molecular mechanism of morphological changes in the OE lines, we performed RNA-seq analysis of the shoot apical meristem (SAM). A total of 2677 genes (24% of all genes differentially expressed in either OE line) were identified as core differentially expressed genes (DEGs) in *GhKTI12* transgenic tobacco (Appendix A). Among those, 518 and 406 genes were upregulated and downregulated, respectively (Appendix A). Analysis of the core DEGs showed that some plant development-related genes were upregulated and some flowering-related genes were downregulated in *GhKTI1*2 transgenic plants. These genes were enriched in 40 GO functional groups, including cell components, molecular functions, and biological processes.

### 4.1. Overexpression of GhKTI12 Could Upregulate Cellular Components Related Genes to Promote Plant Growth

Among the GO enriched cellular components genes in *GhKTI12* transgenic tobacco plants, nineteen genes were significantly upregulated and related to plant development which belong to respiratory chain, microtubule organizing center part, lysosome and α-L-fucosidase activity (Figure 6b).

The respiratory chain-associated genes participate in cell-to-cell communication processes during cell morphogenesis, motility, and fusion in plants. These genes mainly regulate growth, division, differentiation, movement, and respiratory maintenance of cells, as well as cell wall stability in plant development [31], such as *AOX*, *TET*, *CYTC*, *COX*, *F1F0* et al. *AOX* could interact with other cellular signals [32,33] to affect plant biomass production [34], plant growth and photosynthesis rates [35,36], and biotic/abiotic stress tolerance [37,38,39]. *TET8* could regulate plant cell differentiation and maintain the cell membrane microenvironment [40], *TET5* and *TET6* could negatively regulate plant cell proliferation to affect leaf shape and root growth [41]. *CYTC-2* and *COX* could regulate cell respiration and apoptosis [42]. *F1F0* regulates cell proliferation and survival [43]. In addition, the mutation of *CYTC-1* and *CYTC-2* results in smaller rosettes with a significant decrease in parenchymatic cell size and delayed plant development due to the lower mitochondrial respiration rate [44]. These upregulated genes affected cell development in *GhKTI12* transgenic tobacco plants to grow in a bigger morphology and higher yield.

The microtubule organizing center part related genes directly regulated cell development, such as *KTN80*, *CP1* and *KIC*. *KTN80* belongs to the WD-repeat protein superfamily, which is involved in the rapid reorganization of cellular microtubule arrays [45] to regulate cell division and motility [46]. ktn80 mutant Arabidopsis leaves display abnormal epidermal cells due to intertwined and twisted microtube formation [45]. *CP1* and *KIC* are Ca^2+^-binding proteins involved in cell division by interacting with Kinesin-like calmodulin binding protein (KCBP) [47,48,49]. Overexpression of *KIC* in Arabidopsis could regulate leaf trichome morphogenesis through facilitating the microtube-stimulated KCBP protein [50]. Otherwise, increased cellular Ca^2+^ levels could promote primary root growth by mediating auxin signaling in *KIC* transgenic plants [51,52,53]. So the reason of bigger morphology in *GhKTI12* transgenic tobacco lines maybe caused by the upregulation of microtubule organizing center part related genes (Figure 6a and Appendix A). Meanwhile, larger root systems in *GhKTI12* transgenic tobacco lines may be due to upregulation of *CP1* and *KIC*.

In the lysosome, the cathepsins are cysteine proteases of papain-like *C1A* involved in multiple physiological processes such as plant growth seed germination, another development, programed cell death (PCD), abiotic stress, and immunity [54]. In our *GhKTI12* transgenic tobacco lines, five cathepsin proteins genes were upregulated, such as *Cpr-5*, *SAG12-2*, *SEN102*, *AT4G01610* and *RD21A*. These genes could regulate leaf senescence and plant growth. For example, the senescence-specific cysteine protease gene *SAG12* negatively regulates leaf cell death. The downregulation of *OsSAG12-1* resulted in stress-induced cell death in transgenic rice RNAi lines [55]. Several studies have also reported that the expression of a cytokinin biosynthetic enzyme under the control of the *AtSAG12* promoter delays senescence in Arabidopsis, lettuce, and wheat [56,57,58]. In our study, the chlorophyll content in upper leaves (12th leaf) was higher in the OE lines than in the WT, consequently delaying leaf senescence in transgenic plants. It has been also noted that the total amount of carbon fixation in crops increases with prolonged periods of photosynthesis and leaf senescence delay [59]. Therefore, a delay in leaf senescence with increased chlorophyll content in transgenic leaves may be due to the upregulation of *NtSAG12* protein in *GhKTI12* transgenic tobacco plants. A prolonged life stage with increased chlorophyll and a delay in leaf senescence could increase the photosynthetic rate and promote plant development in tobacco overexpressing *GhKTI12*.

Combined gene expression profiles with morphology data demonstrated that overexpression of *GhKTI12* in tobacco could positively regulate cellular component-related genes to promote cell proliferation and plant development. We propose that these upregulated genes promote plants’ development of larger leaves, increased plant height, and larger root systems compared to the WT.

### 4.2. Overexpression of GhKTI12 Could Downregulate Flowering Related Genes to Delay Tobacco Flowering Time

OE-5, OE-6 and OE-8 plants displayed late flowering, delayed by 28–30 days compared with the WT (Figure 2d). This is consistent with Arabidopsis elp1 knockout mutants and ELP1 overexpression plants [2,3,14]. In order to explore the underlying molecular mechanism of delayed flowering in *GhKTI12* transgenic tobacco plants, 10 downregulated genes were identified by RNA-seq analysis in OE-6 and OE-8 lines. Eight of the downregulated genes were MADS-box transcription factor family proteins involved in flowering time regulation, such as *AP1*, *AP3*, *AGL6*, *AGL8*, *AGL14*, *SEP* et al. The gene expression pattern was validated by qRT-PCR (Figure 7b and Appendix A).

The *AP1* and *AP3* can act either alone or in combination to determine the activation of the floral organ-specific gene in specific regions of the developing flower [60,61]. *AP1* plays a crucial role in the transition from vegetative to reproductive phase and floral development [62], and activates the expression of the B-class homeotic genes. *AP3* determines the identity of petals and stamens [63]. Overexpression of AP1 causes early-flowering phenotypes in the Arabidopsis [64], apple [65] and soybean [66].

Furthermore, *AGL6*, *AGL8* and *AGL14* are pivotal in promoting floral meristem development by limiting stem cell proliferation [67,68]. AGL6 has a similar function to *AP1* and *SEP* in Arabidopsis [69]. Accordingly, the overexpression of *AGL6* in Arabidopsis and overexpression of the *SEP-like* gene *TaMADS1* in wheat [70] exhibited early flowering in transgenic plants. Recent studies also demonstrated that the overexpression of *xal2/AGL14* causes early flowering, whereas a knockout mutant shows late flowering in Arabidopsis [71].

These results proved that a delayed flowering time and longer vegetative stage in *GhKTI12* transgenic plants were caused by the downregulation of *MADS-box* genes such as *AP1*, *AGL6*, *AGL14*, and *SEP* genes. Based on the molecular mechanism of morphology and later flowering phenotype in transgenic plants, we hypothesize that the overexpression of *GhKTI12* in tobacco could create a strong basis for plant reproduction and yield by increasing plant size and delaying the flowering time with a longer vegetative stage.

### 4.3. Overexpression of GhKTI12 Could Increase Plants Biomass and Seed Yield

Biomass and seed yield were significantly increased in *GhKTI12* OE lines. The average dry weight of leaves and stems increased by ~34% in OE-5, OE-6 and OE-8 (Table 1 and Appendix A), and the average fresh and dry weight of roots were increased by 33% and 52%, respectively, in both OE-5, OE-6 and OE-8 (Table 1 and Appendix A). These results suggest that the overexpression of *GhKTI12* promotes whole plant development to increase biomass. Transgenic plants additionally produced more flowers and capsules; the average capsule number per ten plants in OE-5, OE-6 and OE-8 was 114, 128 and 142, contributing to an increase of 35%, 40% and 44% net seed yield, respectively (Figure 4d,e). However, there was no significant difference in the average weight of 1000 seeds between lines (Appendix A).

As the main product of photosynthesis, sucrose is transported from the source organ to the sink organ in plants and is related to biomass and yield [72]. We found increased chlorophyll content in younger leaves at the top canopy of transgenic plants and a corresponding decrease in the accumulation of sucrose and starch (Figure 5b,c). This may be due to photosynthetic carbon partitioning between sucrose and starch in the cytoplasm and chloroplasts of leaves, respectively [73,74,75]. Therefore, the overexpression of *GhKTI12* could increase starch accumulation via sucrose breakdown in younger leaves of the top canopy before floral transition, which may contribute to an improved photosynthetic rate, and in turn to an increased seed yield.

Furthermore, the cellular component-related protein *RD21* physically interacts with water-soluble chlorophyll proteins, which are involved in embryo development and seed formation via controlling cell death during the flower development progress [76]. Therefore, high yield in *GhKTI12* overexpressing tobacco may also be caused by the upregulation of *RD21*. Taken together, these results suggest that the overexpression of *GhKTI12* could improve seed yield and biomass production by regulating gene expression involved in various biological processes including cell division, leaf senescence, programed cell death, photosynthesis, and embryo development.

## 5. Conclusions

In conclusion, the overexpression of *GhKTI12* in tobacco led to significant variation in agriculturally important phenotypes, increasing cell proliferation and organ growth, delaying flowering time, and increasing plant height, seed yield, and biomass production. The OE-5, OE-6 and OE-8 lines with significantly improved phenotypes illustrate that *GhKTI12* acts as a super-gene to positively regulate plant development and yield. Thus, *GhKTI12* has broad prospects for the genetic improvement of crops in the future.

## Figures and Tables

**Figure 1 genes-13-00426-f001:**
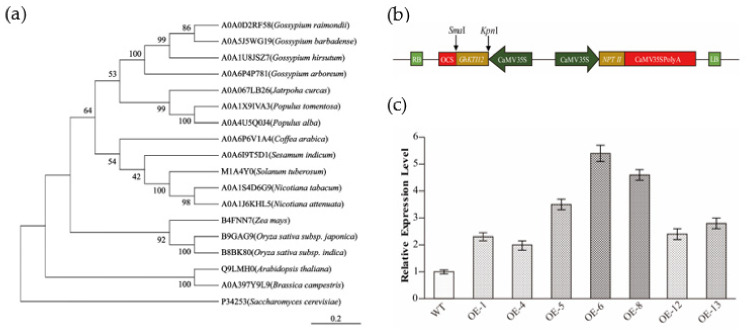
Phylogenetic analysis of GhKTI12 homologs and identification of transgenic plants. (**a**) Phylogenetic analysis of GhKTI12 homologs in plants. NJ tree is generated by MEGA-X software using full protein sequences. Numbers in each branch indicate distance genetic variation between species. GhKTI12 was shown as A0A1U8JSZ7 (upland cotton). (**b**) Vector constructions of *GhKTI12* based on PCAMBIA2300-35S backbone. (**c**) qRT-PCR analysis of relative transcript level of *GhKTI12* in T0 transgenic tobacco lines (Line 1–7). Bars in the graph show the standard mean error. Asterisks indicates significant differences between wild type (WT) and *GhKTI12* transgenic plants analyzed by Student’s *t*-test, *p* < 0.01.

**Figure 2 genes-13-00426-f002:**
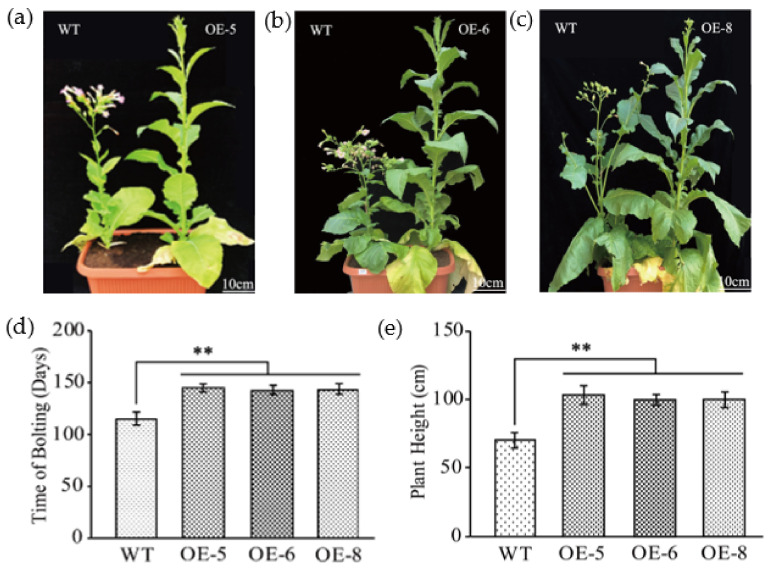
Flowering time and plant height analysis in *GhKTI12* transgenic plants. (**a**–**c**) Plant height (Scale = 10 cm) and initial bolting morphology of WT and *GhKTI12* transgenic tobacco. Statistical comparison of morphological analysis in (**d**) Initial bolting time. (**e**) Plant height (Scale = 10 cm). Bars in the graph show the standard mean error. Asterisks indicates significant differences between wild type (WT) and *GhKTI12* transgenic plants analyzed by Student’s *t*-test, *p* < 0.01 (**). Data were collected from 10 representative plants.

**Figure 3 genes-13-00426-f003:**
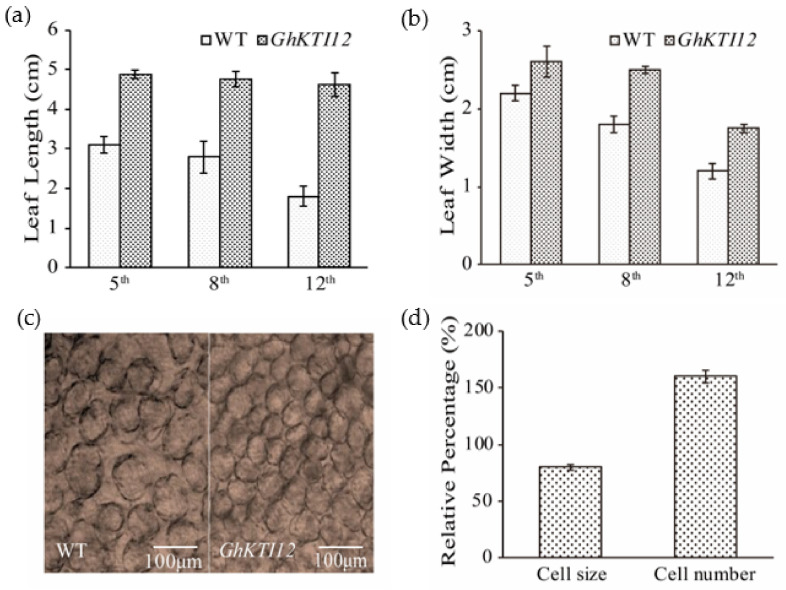
Effect of *GhKTI12* on tobacco leaf development. (**a**,**b**) Comparison of leaves size from three different positions (5th, 8th, and 12th) between WT and *GhKTI12* transgenic plants from 4-month-old plants, the leaf length and leaf width were measured (n = 10). (**c**) Microscopic observation of palisade cell in the upper epidermis of third leaf in wild type and transgenic plant. (Bars = 100 um.) (n = 5). (**d**) Relative percentage of epidermis palisade cell number and cell size in *GhKTI12* transgenic tobacco compared with wild type (n = 10). All data was taken from T2 generation. Bars in the graph show the standard mean error. Asterisks indicates significant differences between wild type (WT) and *GhKTI12* transgenic plants analyzed by Student’s *t*-test, *p* < 0.01.

**Figure 4 genes-13-00426-f004:**
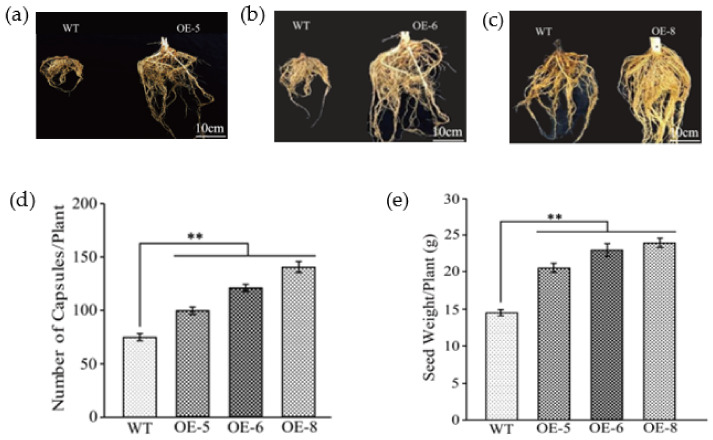
Analysis of root morphological and seed production in transgenic plants. (**a**–**c**) Comparison of roots morphological between wild type and *GhKTI12* transgenic plants. (**d**) Total number of capsules per each plant (n = 10). (**e**) Seed yield per individual tobacco plant (n = 10). Bars in the graph show the standard mean error. Asterisks indicates significant differences between wild type (WT) and *GhKTI12* transgenic plants analyzed by Student’s *t*-test *p* < 0.01 (**). Data were collected from 10 representative plants.

**Figure 5 genes-13-00426-f005:**
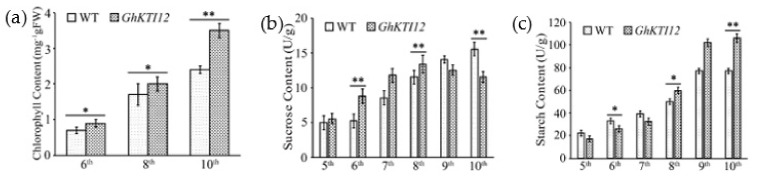
Contents analysis of chlorophyll, sucrose and starch in *GhKTI12* transgenic tobacco leaves. (**a**) Chlorophyll content in *GhKTI12* transgenic plants and wild-type plants (n = 10). (**b**) Sucrose content in leaves at different leaf ages. (**c**) Starch content in leaves at different leaf ages. All data were taken from T2 generation. Bars in the graph show the standard mean error. Asterisks indicates significant differences between wild type (WT) and *GhKTI12* transgenic plants analyzed by Student’s *t*-test, (*) *p* < 0.05; (**) *p* < 0.01.

**Figure 6 genes-13-00426-f006:**
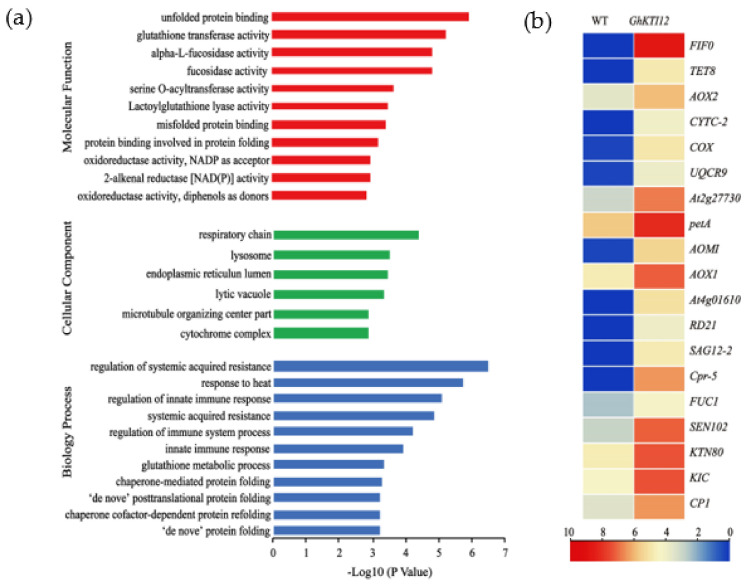
Analysis of upregulated genes in *GhKTI12* transgenic plants. GO Histogram of Annotated (**a**) upregulated genes (URGs) in three different functional categories. (**b**) The heat map of DEGs associated with enriched cellular components.

**Figure 7 genes-13-00426-f007:**
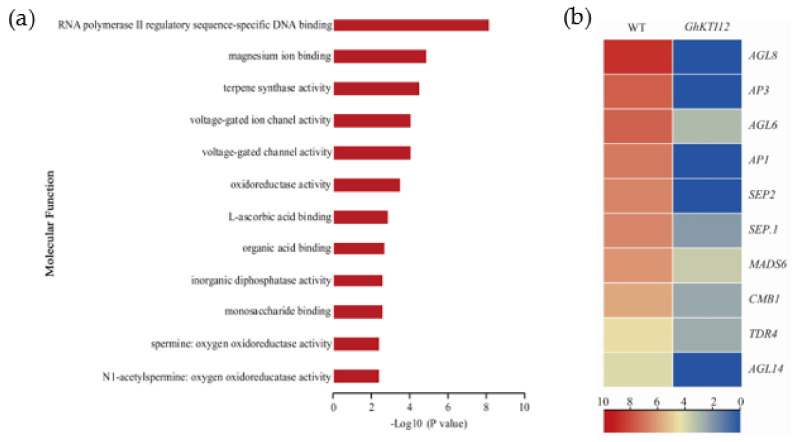
Analysis of downregulated gene related to plant flowering in GhKTI12 transgenic plants. (**a**) Top 12 GO enrichment analysis of downregulated genes (DRGs) in molecular function. (**b**) The heat map of DGEs associated with flowering time related genes.

**Table 1 genes-13-00426-t001:** Comparison of biomass production between *GhKTI12* transgenic and wild type (WT) tobacco plants.

Genotype	Fresh Weight (g)				Dry Weight (g)			
	Leaves	Stem	Root	Total Weight (g)	Leaves	Stem	Root	Total Weight (g)
WT	110.66 ± 9.11	148.73 ± 14.23	20.47 ± 2.03	279.86	26.05 ± 2.11	29.93 ± 2.06	3.76 ± 0.71	59.74
OE-5	173.15 ± 13.18	232.58 ± 15.02	32.01 ± 2.92	437.74	40.74 ± 3.98	43.26 ± 3.80	8.26 ± 0.94	92.26
OE-6	163.93 ± 15.16	220.19 ± 17.81	30.3 ± 2.82	414.42	38.57 ± 3.56	43.46 ± 3.72	7.82 ± 0.72	89.85
OE-8	165.01 ± 14.48	221.64 ± 17.67	30.52 ± 2.65	417.17	38.82 ± 3.64	44.42 ± 3.17	7.87 ± 0.97	91.11

Data were collected at harvesting time. Standard deviation (SD) were calculated from mean value of 10 sample plants.

## Data Availability

The datasets generated during and/or analyzed during the current study are available from the corresponding author on reasonable request.

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
