# Peer review of "Overexpression of GhKTI12 Enhances Seed Yield and Biomass Production in Nicotiana Tabacum"

_genes, 2022, doi:10.3390/genes13030426_

Round 1

Reviewer 1 Report

Authors performed generally interesting and valuable research related to heterologic expression of KTI12 in N. tabacum.

Study is well planned and performed. Authors obtained novel information related to morphological, histological, biochemical (glucose and starch content) and gene expression analysis that is associated with overexpression of GhKTI12 in M. tabacum. Transgenic plants indicated delayed flowering time and longer vegetative growth phase. Authors identified by transcriptomic studies and then validated by RT-PCR analysis a group of genes that are up- and down-regulated in transgenic plants. These genes may be responsible for observed morphological and physiological changes.

Some corrections are suggested to further improve the manuscript:

Latin names of plants or other organisms in the entire text should be written in italics- lines 42, 46 and other.

Introduction

Authors should add to this section several sentences of the Elongator complex, its structure, association with RNA polymerase II, role in transcription regulation. It is generally stated in the introduction but is should be more clearly shown.

Section 2.2

How the genomic DNA was isolated, how the homozygous state of T2 lines was assured?

Usually homozygotic lines are developed by several (5-6) cycles of self -crossing.

Concentration of kanamycin used for transgenic plants selection, duration of kanamycin selection.

Polymerase (name, manufacturer) used to amplify the GhKTI12. Amount of gDNA used as a template in the PCR.

Section 2.3

How the quality of RNA was assured?

Amount of RNA taken for analysis.

Details of RT reaction.

How the remnants of genomic DNA were removed?

Details of PCR reaction.

Length of PCR products- tested and reference gene

Previous use of reference gene- citation or stability analysis using Bestkeeper or related software.

Name and manufacturer of RT-PCR equipment.

Software used for analysis of raw RT-PCR results.

Section 2.2, 2.4 and others

There should be no space between number for example 65 and ⁰C. It should be 65⁰C but not 65 ⁰C. Correct in the entire text.

Section 2.8

How the quality of RNA was assured?

Volume and concentration of cDNA library.

How Authors managed with  the error rate of PacBio sequencers?

What quality parameters characterized obtained sequences?

What units of relative gene expression were used RPKM or other?

Was the completeness of transcriptome analysed using for example Busco software?

Section 3.2

What was the difference between three lines overexpressing the GhKTI12?

Reviewer 2 Report

The main aim of the research is the discovery of the GhKTI12 gene influence on Nicotiana tabacum growth and mechanisms of its influence on yield, vegetative period and chemical composition. After N. tabacum transformation many different analyses of regenerated plants (morphological description, histological and biochemical analysis, RNA sequencing) were carried out. All these helped to understand the mechanisms of GhKTI12 gene expression. Finally the analyses of 3 transformed lines showed that GhKTI12 gene positively super regulates plants development and yield.

Remark:

Line 85. Species name Gossypium hirsutum L. as a botanical name must be printed in italics.

Some English grammar revision is needed.
